# Multifunctional and Self-Healable Intelligent Hydrogels for Cancer Drug Delivery and Promoting Tissue Regeneration In Vivo

**DOI:** 10.3390/polym13162680

**Published:** 2021-08-11

**Authors:** Elham Pishavar, Fatemeh Khosravi, Mahshid Naserifar, Erfan Rezvani Ghomi, Hongrong Luo, Barbara Zavan, Amelia Seifalian, Seeram Ramakrishna

**Affiliations:** 1Pharmaceutical Technology Institute, Mashhad University of Medical Sciences, Mashhad 91735, Iran; mahshidnaserifar@yahoo.com; 2Center for Nanotechnology and Sustainability, Department of Mechanical Engineering, National University of Singapore, Singapore 117581, Singapore; fatemeh_khosravi22@yahoo.com; 3Engineering Research Center in Biomaterials, Sichuan University, Chengdu 610064, China; hluo@scu.edu.cn; 4Department of Morphology, Experimental Medicine and Surgery, University of Ferrara, Via Fossato di Mortara 70, 44121 Ferrara, Italy; zvnbbr@unife.it; 5UCL Medical School, University College London, London WC1E 6BT, UK; asseifalian@yahoo.co.uk

**Keywords:** anti-cancer drug delivery, regenerative medicine, self-healing hydrogel, cancer vaccination

## Abstract

Regenerative medicine seeks to assess how materials fundamentally affect cellular functions to improve retaining, restoring, and revitalizing damaged tissues and cancer therapy. As potential candidates in regenerative medicine, hydrogels have attracted much attention due to mimicking of native cell-extracellular matrix (ECM) in cell biology, tissue engineering, and drug screening over the past two decades. In addition, hydrogels with a high capacity for drug loading and sustained release profile are applicable in drug delivery systems. Recently, self-healing supramolecular hydrogels, as a novel class of biomaterials, are being used in preclinical trials with benefits such as biocompatibility, native tissue mimicry, and injectability via a reversible crosslink. Meanwhile, the localized therapeutics agent delivery is beneficial due to the ability to deliver more doses of therapeutic agents to the targeted site and the ability to overcome post-surgical complications, inflammation, and infections. These highly potential materials can help address the limitations of current drug delivery systems and the high clinical demand for customized drug release systems. To this aim, the current review presents the state-of-the-art progress of multifunctional and self-healable hydrogels for a broad range of applications in cancer therapy, tissue engineering, and regenerative medicine.

## 1. Introduction

Tissue engineering offers alternative ways to address the current challenges associated with autologous grafts by developing highly porous biomaterials and scaffolds to encapsulate cells that spread and reorganize into tissue-like architectures to replace damaged tissues [1,2]. Among many scaffolding biomaterials, hydrogels are considered as a unique group of three-dimensional (3D) polymeric substances due to their biomimetic properties, biocompatibility, porous structure, and the capability to absorb and retain a high amount of water, and adaptability using interchangeable sol-gel conditions [3,4]. Hydrogels are widely utilized in various healthcare applications such as wound dressings, contact lenses, sensors, and drug delivery systems [5]. The most common water-soluble polymers include poly(vinylpyrrolidone), poly(acrylic acid), poly(vinyl alcohol), poly(ethylene glycol), polyacrylamide, and some polysaccharides. Many hydrogels have been patented for drug delivery like SUPPRELIN LA, which is utilized for the treatment of children with central precocious puberty (CPP) through sustained release histrelin acetate [6]. Although hydrogels have many useful properties, traditional hydrogels are often not ideal for tissue engineering applications due to their irreversibility of covalent crosslinks, leading to being easily broken during normal operation [6,7]. Since conventional hydrogel-based contact lenses reveal fairly low drug-loading ability and often lead to burst release drugs for ocular administration, some factors such as control hydrophilic/hydrophobic copolymer ratio, impregnating drug-containing colloidal structures, incorporating ligand-including hydrogels, and developing multilayered hydrogels improve release drugs. Incorporation hydrogel poly-2-hydroxyethylmethacrylate (PHEMA) with b-cyclodextrin (beta-CD) hydrogel in contact lenses exhibited longer residence time of H1-antihistamines compared to conventional PHEMA hydrogel via enhancing swelling ratio and tensile strength [8]. Furthermore, natural hydrogels for biosensor applications could overcome the limitation of traditional polymers due to biocompatibility, their inherent renewable, nontoxic features, and biodegradability [9]. Soft hydrogel based self-healing triboelectric nanogenerator (HS-TENG) consist of photothermic-active polydopamine nanoparticle and multiwalled carbon nanotube (MWCNT) allows inside poly(vinyl alcohol)/agarose hydrogel to enhance mechanical and electrical property for wearable applications. This self-healing hydrogel is improved by NIR light and the exposure of water spray [10]. TC-Gel sensor as an electronic skin device was prepared from polyaniline (PANI) network and covalently cross-linked poly (acrylic acid) (PAA) network, in which tannic acid-coated cellulose nanocrystals (TA@CNCs) was integrated inside network as reinforcing physical cross-linkers to obtain superior mechanical and electrical properties. The dynamic hydrogen bonds TA@CNCs lead to electrostatic interactions in the hydrogel networks to be easily reformed at room temperature after being broken and providing excellent autonomous self-healing capability and high-stability contact on human skin [11]. Meanwhile, a coating implant with conductive hydrogels led to better interfaces with neural tissues. For example, polyacrylamide and poly(3,4-ethylenedioxythiophene) polystyrene sulfonate have revealed the excellent adhesion to implant (polydimethylsiloxane), which is commonly used as a substrate in soft neural interfaces [12]. Numerous different applications of natural self-healing hydrogels have been organized in Table 1, which is available in marketing.

In other words, the application of various injectable hydrogels is limited by control gelation time. Slow gelation could give rise to rapid material dispersion and loss of cargo, while clogging causes rapid crosslinking during the injection device [13]. Many factors are involved in communication between hydrogels and stem cells, affecting stem cell survival, such as polymer types, stiffness, porosity, degradation, and compatibility [14,15]. The shape of smart hydrogels could alter under the effect of an appropriate stimulus because of reversible covalent or physical cross-links [16]. Combination of hydrogels with flexible three-dimensional structures and self-healing property has been made into novel intelligent hydrogels [17]. Recently, intelligent hydrogels have been investigated with self-healing properties, which means the capability of the material to recover its initial structure after being damaged [18,19]. This phenomenon is like the same healing process of natural organisms via the reversibility of dynamic covalent crosslinks without an external stimulus, e.g., low pH, enzymatic environment, electric fields, temperatures, and UV light [20]. These hydrogels mimic the 3D ECM to regenerate after minor injuries similar to native tissues. In other words, self-healing capability refers to a gel’s ability to recover to its original shape via noncovalent interactions, dynamic covalent, and physical bonds (Figure 1) [21]. Supramolecular hydrogels consist of self-assembling peptides, hosts such as cyclodextrins and polysaccharides, which could improve interactions between two types of different polymers and give rise to higher self-healing properties [22]. Self-healing hydrogels have beneficial properties, including injection via needles without clogging, homogenous encapsulation of payloads, recovery to their initial state, and easy delivery of damaged tissue [13].

Furthermore, intelligent hydrogels have great potential for targeted drug delivery, non-invasive, remote-controlled therapies, regenerative medicine, tissue engineering, and implanting artificial organs. Dynamic covalent bonds show stable and slow dynamic equilibriums, whereas noncovalent interactions indicate rapid dynamic equilibriums and fragile ones [23]. However, despite the recovering feature of self-healing hydrogels, their application is restricted due to poor conductivity and lower fracture energies than that of native tissue [23]. One approach to overcome the limitations of intelligent hydrogels is a combination of synthetic and natural polymers with nanomaterials to enhance biocompatibility and physical capability [14].

The local delivery of therapeutic agents and drugs are currently utilized for inflammation prevention, cancer elimination, and the regeneration of injured tissue [24]. Noticeably, the local delivery can decrease the limited toxic effect of a drug in the tumor area without any cytotoxic effect for the adjacent healthy tissues. Self-healing hydrogels suggested avoiding a chronic foreign-body immune response after surgery. Besides, they could release drugs for a long time and be stimuli-responsive for an on-demand drug [1]. The structure of intelligent hydrogels can be altered by different types of stimuli from internal (e.g., in-body) or external (i.e., out-body) sources. Internal and external sources are classified as biological, physical, and chemical [25].

This manuscript aims to summarize important studies conducted toward developing self-healing hydrogels for various applications in regenerative medicine and cancer delivery applications. We investigated supermolecular hydrogels with dynamic bonds that possess mechanical and conductive properties for cell and drug delivery. In this regard, the current main challenges and issues to apply hydrogels as scaffolds for drug or biologic agent delivery to maintain, regenerate, and modify lost or damaged tissues are discussed and summarized.

## 2. Tissue Engineering Applications and Cancer Drug Delivery

Several properties of the synthetic and natural hydrogels, such as biocompatibility, biodegradability, mechanical properties (applicable to the bone and muscle tissue), and electrical feature (applicable to cardiac and nerve tissues), should come into account for better integration with native organs [26,27]. Furthermore, scaffolds should trigger cell attachment, proliferation, and differentiation without a cytotoxic and minimal immune reaction [28]. Numerous studies have focused on the development of self-healing hydrogels that spontaneously possess electroactivity with stable mechanical properties [26]. The dynamic nature of reversible crosslink facilitates recapitulating the viscoelastic nature of the ECM and leads to strengthening restoring behavior and shear-thinning (decline in viscosity as shear stress rises) behavior, which is essential for injection [27]. Figure 2 summarizes the main elements in tissue engineering based on engineering stem cells, nanoparticles, and novel self-healing polymers to enhance the structural and functional resemblance to the tissues. Recent investigations have tried to modify mesenchymal stem cells (MSCs) by incorporating beneficial genes/drugs to augment the differentiation capacity or induce the production and secretion of growth factors [29,30,31], which is beyond the scope of this article. This section will consider recent developments of self-healing hydrogels focusing on cancer drug delivery, bone, cardiac, neural, and lung tissue engineering.

### 2.1. Cancer Drug Delivery

Cancer is considered as the second global cause of death [32,33]. Due to the lack of reliable treatment alternatives and high systemic doses of chemotherapy and drug resistance [26], matrices with features of sustained drug release are crucial [34]. Smart self-healing injectable hydrogels provide sustained release of bioactive agents, such as micro- or macromolecular drugs [35]. Natural hydrogels have widely been used as drug delivery vehicles such as HA, collagen, gelatin, alginate, chitosan [36]. Hydrogels based on different parameters can be classified based on network size (macrogels, microgels, nanogels), composition (homo or copolymeric), electrical charge (non-ionic, cationic, anionic, amphoteric, or zwitterionic), crosslinking (physical or chemical) [37]. Some hydrogels can provide a response to an environmental stimulus such as ionic force or pressure, temperature, pH, and light which reflect to their structure [37]. 

Self-healing hydrogels offer benefits compared with conventional hydrogels for drug delivery, such as homogeneous encapsulation of loaded drugs [38], preventing drug diffusion, improving the treatment effect of drugs, decreasing toxicity to normal tissues [33], reducing viscosity through enhancing the rate of shear stress and resistance to the stress-induced formation of cracks [39], which result in lingering for their lifetime during injection [40]. Here, we report some sensitive hydrogels with self-healing ability to control cancer drug release (Figure 3).

PH-sensitive self-healing polysaccharide-based hydrogels were prepared as Doxorubicin (Dox) drug delivery vehicles for hepatocellular carcinoma therapy by dynamic covalent Schiff-base linkage between amine groups from carboxyethyl chitosan (CEC) and benzaldehyde groups from dibenzaldehyde-terminated poly (ethylene glycol) (PEGDA). Since chitosan is biodegradable, nontoxic, and biocompatible, it is used for extensive applications in clinical fields. Release of Dox from CEC/PEGDA hydrogel was evaluated at pH 4, 5.5, and 7.4. The rate of drug release at pH 4.0 was more rapid than others with approximately 96% during four days. In addition, 92 and 89% of Dox were released in PBS at pH 5.5 and 6.8 after seven days of incubation, respectively. On the other hand, 42% of Dox was released in PBS at pH 7.4 after seven days. Furthermore, Their results showed that Dox concentration plays a crucial role in both burst and cumulative release ratio [35]. In another study, Liposomes/curcomin was encapsulated by thiolated chitosan (CSSH), which was fluidic at room temperature and gelled quickly at 37 °C. CSSH/Cur-Lip gel leads to the cumulative release of curcomin at approximately 31% at 12 h in breast cancer treatment [41]. Protein-based hydrogel has also been utilized as prominent scaffolds in various biomedical applications such as delivery, sensing, tissue engineering, and fabricating artificial organs due to their biocompatibility, biodegradability, viscoelasticity in the body [42]. For example, Upadhyay et al. utilized bovine serum albumin (BSA) as a scaffold to promote gelation by using glutaraldehyde as an external crosslinking agent or by internal crosslinking through a disulfide bridge. The release of Dox from BSA hydrogel occurred over five days to the extent of almost 37, 26, and 21% in PBS at pH 5.5, 6.8, and 7.4, respectively, which result in mortality of three cancer cells (MCF-7, HeLa, and MDA-MB-231) [43]. Yang et al. used chitosan derivatives dynamic network consisting of glycol-chitosan crosslinked by the benzaldehyde to form Schiff base to steadily deliver a highly concentrated antitumor drug (Taxol) for seven days for treatment of human hepatocarcinoma tumor [44].

Nowadays, photothermal therapy (PTT) plays a crucial role in treating cancers due to its low toxicity and high efficiency [45]. The fabrication of self-healing hydrogels is promoting an approach to sustain drug release through the hydrophilic–hydrophobic connection. In other words, some nanomaterials, including gold, carbon nanotubes, graphene, and Fe_3_O_4,_ can be activated with near-infrared (NIR) light, which can lead to producing thermoresponsive self-healing hydrogel [46]. Among them, polydopamine (PDA) is the most efficient due to its distinctive adhesion, biocompatibility, and well dispersion in hydrogels to produce heat to change the morphology of hydrogels [47]. Wang et al. fabricated a thermoresponsive self-healing hydrogel through dynamic covalent enamine bonds between polyetherimide (PEI) and poly(2-(dimethylamino)ethyl methacrylate) (PDMAEMA) copolymer. They used the PDA (0.2 wt.%) in the hydrogel to control the release of Dox under NIR laser irradiation. Subsequently, the construction of hydrogel changed from hydrophilic to hydrophobic. PDA concentration in the hydrogel is important because of its effect on strength and injectability [48].

Another thermoresponsive self-healing hydrogel composed of graphene conjugated to branched polyethyleneimine (BPEI) was prepared to interact with chondroitin sulfate multialdehyde (CSMA) hydrogel [49]. BPEI-GO was dispersed in CSMA to make sustained drug delivery with a photothermal effect [49]. Graphene oxide (GO) was also used in this study due to its mechanical strength, tumor therapy, surface modifiability, good colloidal stability, biocompatibility, and near-infrared (NIR) absorption, which transforms absorbed light into heat. The results indicated that more Dox could release from BPEIGO at pH 6.5 (57.05%) compared to pH 7.4 (33.89%) and pH 10.0 (36.73%) in 24 h with CSMA 30 wt.% [49]. Furthermore, codelivery has shown several benefits through synergistic effects of different agents (drug/drug or drug/gene) with reduced side effects property [50]. One of the limitations of hydrogel codelivery drugs is maintaining the sequential and sustained release of multiple drugs [51]. Yavvari et al. used catechol-based hydrogel chitosan (CAT-Gel) as injectable and self-healing hydrogel, which assembled via catechol-Fe(III) coordinative connections for local delivery hydrochloride (Dox) and hydrophobic docetaxel (DTX) anti-cancer drugs in murine lung and breast cancer models. Their result demonstrated that Dox release was faster than hydrogel in 20 days with an antitumor effect, whereas DTX released slowly due to interactions of DTX with hydrogel network, as shown in the rheological studies [51].

Notably, nanogels have advantages for drug delivery, including easily modifying targeting ligands like aptamers, blood circulation, and reducing the drug dosage to strive for specific recognition of the target cells. For example, nucleoside analogues like floxuridine (F) as cytotoxic drugs have been loaded in DNA and RNA spherical nanogels via solid-phase synthesis or enzyme-mediated transcription [52]. Zhao et al. proposed ATP responsive DNA nanogel for the delivery of DOX to cancer cells. This self-healing core-shell is composed of DNA chains (DNA1 and DNA2), which are rich in guanine (G) and cytosine (C) bases with a carboxyl group at the 5′ terminal of both single-stranded DNA chains to react with amide biocompatible carboxymethyl chitosan (CMCS). Therefore, CMCS-DNA1 and CMCS-DNA2 could use hybrid ATP aptamer as a crosslinking agent to form core-shell NGs NGs@DOX. The results revealed the effect of ATP on the release of DOX from NGs@DOX, whose fluorescence intensity of DOX-loaded NGs was effectively enhanced with higher concentration of 0.3 Mm of the ATP. Besides, pH values (6.5 and 7.4) play an essential role in DOX release from NGs@DOX. It was indicated the fluorescence intensity of DOX released from NGs@DOX in pH 6.5 PBS was higher than that of in pH 7.4 PBS, under the same amount of ATP triggers. Temperature is considered another important factor in releasing DOX from NGs@DOX. Although there was no significant release increase of DOX in the A549 tumor cells at 4 °C, the obvious fluorescence intensity could be identified at 37 °C. It inhibited ATP by (iodoacetic acid, IAA) reagent to confirm the role of ATP in the release of DOX from the hydrogel. It was shown that the capability of uptake of NGs@DOX at 37 °C after incubation during 1, 2, 3, and 4 h on A549 cells. The fluorescence intensity of NGs@DOX-treated A549 cells was slightly more increased than that of free DOX after 2 h incubation. Notably, cell viabilities were more than 80% in the group treated with NGs without DOX, which revealed the suitable biocompatibility of the carriers. DOX-loaded nanogels displayed lower viabilities than free DOX due to the better cellular uptake of NGs@DOX and ATP-responsive release DOX in cancer cells [53].

Cancer immunotherapy has progressed immediately due to target and durable antitumor responses via the use of the host’s natural immune system. Leach et al. suggested peptide hydrogel for delivery synthetic cyclic dinucleotides (CDNs) as stimulating strong antitumor responses in preclinical models. Self-assembling peptide nanofiber hydrogel with sequence K_2_(SL)_6_K_2_ (MDP) illustrated eight-fold slower release rate CDNs compared to a standard collagen hydrogel, thereby reducing infiltration in a normal cell in local delivery model of head and neck cancer tumors in wild type C57BL/6 mice with a single injection of hydrogel [54]. Furthermore, gel-based could promote retention of immunotherapeutics like antibodies in a tumor. Sprayed immunotherapeutic fibrin gel composed of fibrinogen and thrombin with CaCO_3_ nanoparticle to deliver aCD47 and control release in the acidic tumor microenvironment during six days in mouse animal model blocking the interaction of CD47 on cancer cells could enhance phagocytic cancer cells by macrophages, dendritic cells, and neutrophils [55]. Thermosensitive hydrogels have evaluated synergistic antitumor efficacy vaccines through regulating immune function and inducing cell cycle arrest based on PEG and poly(γ-ethyl-L-glutamate) formulation for local codelivery of cytokine (IL-15) and cisplatin (CDDP) on C57BL/6 mice bearing melanoma. This hydrogel can release a therapeutic agent during weeks after subcutaneous injection, in which after 2 days of incubation, approximately 66% of drug (CDDP) was released from the hydrogels, compared to IL-5 with 37% released [56]. Another stimulation tumor factor that affects altering shape hydrogel is reactive oxygen species (ROS). Gu et al. used poly(vinyl alcohol) (PVA) with a ROS-labile linker *N*1-(4-boronobenzyl)-*N*3-(4-boronophenyl)-*N*1,*N*1,*N*3,*N*3-tetramethylpropane-1,3-diaminium(TSPBA) to gemcitabine (GEM) delivery and enhanced antitumor responses through local release of anti–PD-L1 blocking antibody (aPDL1) in the B16F10 melanoma and 4T1 breast tumor [57]. Table 2 summarizes the in vitro and in vivo studies for cancer drug delivery using self-healing hydrogels.

### 2.2. Bone

Extracellular matrix (ECM) bone mainly comprises of 0–70% primarily inorganic hydroxyapatite (HAp), 20–40% organic constituents (primarily type I collagen), 5–10% water, and 3% lipid and proteins such as growth factors [25]. Conventional bone treatment methods, including autografts, allografts, and xenografts, are restricted due to their potential infection, risks of donor-site morbidity, high nonunion rate with host tissues, and adverse immune response [58]. There are several factors involved in bone tissue engineering (BTE), including osteogenic scaffold, osteogenic stimulating factors, and osteogenic cells. Hydrogels have been widely used in BTE due to their capability to mimic bone ECM and effective delivery of growth factors, mRNAs, miRNAs, and drug and nanoparticle delivery [59]. Thus, self-healing hydrogel scaffolds have emerged that are biocompatible, nontoxic, and capable of improving cell differentiation and new tissue formation by facilitating the nutrients’ diffusion as a promising alternative to auto and allografts [14,60].

Ideal hydrogels for bone regeneration need to be osteoconductive, osteoinductive, osteocompatible for enhanced bone regeneration, and nonimmunogenic to avoid causing inflammation [61]. These features should be strived for by choice of the hydrogel polymer backbone, crosslinking, and numerous functionalizations. Meanwhile, mechanical strength is considered as one of the vital issues for bone hydrogel scaffolds that need to tackle the load-bearing conditions in the native bone in expediting new tissue ingrowth. A double-network hydrogel was recently used to develop robust hydrogels with high self-healing properties consisting of a strong and rigid polymeric network and catechol groups with reversible and irreversible crosslinks [14]. Notably, the incorporation of nanoparticles and bioglass and ceramics could optimize the physical and chemical properties of hydrogels as well as increasing vascularization. For example, biocomposites contain *β*-TCP with ions Mn^2+^, Zn^2+^, and Sr^2+^ and polymer silk fiber tagger high osteogenesis and cell proliferation, thereby showing the capability of promoting immunomodulation [28].

Other factors required for bone tissue engineering are suitable pore size and interconnected porosity achievable by changing the concentration and variety of polymers, crosslinkers, and fabrication method [62]. These factors will help with improving the controlled release of encapsulated drugs and the exchangeability of oxygen and nutrients in hydrogels [63]. Hence, several studies have been investigated on injectable and 3D printed natural and synthetic hydrogels with a high potential for self-healing [64]. Since crosslinking plays an important role in increasing the mechanical property and stability of hydrogels, we will discuss polymer sources and crosslinking technologies being utilized to construct BTE hydrogels with their advantages and limitations for use in bone repair in this section. 

#### Injectable Hydrogels for Bone Tissue Engineering

Injectability and self-healing properties play critical roles in providing less invasive therapeutic applications for patients. Common approaches to achieving injectable hydrogels include dynamic-covalent bonds, electrical interactions, thermal gelation, imine bonds, Diels–Alder, and host–guest interactions. The reversible nature of crosslinkers should contribute to shear-thinning property hydrogel [65]. Previous reports have mentioned the advantages and drawbacks of natural and synthetic polymers [59]. Commonly utilized self-healing hydrogels are based on prepolymers protein–DNA complexes, HA [66,67], PEG [68], elastin-like polypeptides (ELP) [69], chondroitin sulfate, and silk fifibroin [70]. Backbone polymers such as polypeptide, polyacrylamide, and carbon nanotube can form novel hydrogel via the linker DNA sequences that possess self-healing or self-repairing properties. The advantage of using “X”-shaped DNA as a crosslinker is its responsiveness to temperature, UV, enzyme, pH, and light. For example, Li’s group synthesized the polypeptide-DNA as cross linker to polymer poly(L-glutamic acid-co-γ-propargyl-L-glutamate). A total of 5–6 ssDNAs were conjugated to each polypeptide backbone and sticky ends of the DNA could label with 5(6)-carboxyfluorescein (5(6)-FAM) and 5(6)-carboxy-X-rhodamine (5(6)-ROX) or both of them, which are visualized as green, red and orange, respectively (Figure 4a) [71]. Chain-exchange reactions in DNA double helix were found to be the reason for self-healing with different fluorophore labeling (Figure 4b). The diffusions of DNA chain caused the removal of interfaces among other parts. The next day, the three colorful hydrogels appeared completely and adjusted themselves according to the shape of the container at 4 °C (Figure 4c). Figure 4d exhibited that the mechanical strength of the merged material could spread to 80% of their original value and healing and recovery property of the hydrogel (4 wt.%) that was cut into pieces, and could maintain the original mechanical strength in 5 min [71].

Notably, recent studies have focused on DNA-based hydrogel nanocomposite due to the success of dynamic covalent bonds via imine-based reactions that emerged as a sustained release of drugs. In this regard, Basu et al. utilized nanocomposite DNA-based hydrogels through dynamic reversible imine covalent bonds and the incorporation of silicate-based nanoparticles (nSi) for improving the shear strength of the formulated hydrogel by establishing electrostatic interactions with the phosphate groups of the DNA network. This optimized DNA hydrogel indicated the sustained release of Simvastatin as an osteogenic drug for more than a week [72].

Ureido-pyrimidinone (UPy) hydrogel with the reversible and dynamic quadruple hydrogen bonding capacity has suggested both self-healing and shear-thinning properties for bone–cartilage (osteochondral) interface [39]. UPy hydrogels were developed for drug delivery systems without requiring the incorporation of hydrophobic spacers for gelation. For instance, Hou et al. prepared a self-integrating and injectable polysaccharide supramolecular hydrogel from dextran polymer polysaccharide (DEX) with multiple pendant UPy crosslinking along the DEX backbone for loading bone morphogenetic protein 2 (BMP-2) for supporting the growth of both bone and cartilage tissues in vivo subcutaneous implantation model in nude mice (Figure 5). Three constructions of different kinds of cells were assessed, such as chondrocytes alone, BMSCs/BMP-2, and self-integrated implant with the two types of cells encapsulated on two sides of the gel disk. Markers of osteogenesis and chondrogenesis were evaluated through Alizarin red staining for bone and Alcian blue staining for cartilage, which the formation of cartilage and bone within the single cell-type groups was visualized, respectively (Figure 5A,B). The self-integrated osteochondral implants were formed (Figure 5C), and the image was magnified (Figure 5D). The rheological characterization of the hydrogels depends on the concentration of DEX and UPy. When DEX concentration increased from 10 to 12.5% (*w*/*w*), the storage modulus changed from 170 to 700 Pa, leading to enhance the mechanical performance of the hydrogel [39].

Peptide amphiphiles (PA) hydrogels are considered as self-assembling hydrogels with inherent bioactivity and biocompatibility for 3D cell culture, drug delivery, and applications in hard tissues [73,74,75]. PA is hydrophilic and can attach covalently to other long hydrophobic polymers. PAs have been widely utilized to recreate dentin for delivery of both dental stem cells and growth factors [67,68] through matrix metalloproteinase (MMP) MMP-cleavable linker, which improves viability, migration spreading of human mesenchymal stem cells, and angiogenesis. For example, PuraMatrix™ hydrogel scaffold self-assembling is composed of 16 amino acid peptide sequences called RADA16 with noncovalent β-sheeted structures [76]. The injectable RADA16 hydrogel, when demonstrated successfully, slowed BMP-2 release in vitro. Furthermore, the incorporation of cell adhesion motifs into RADA16 hydrogels enhanced osteoblast cell attachment and migration into the scaffold [76].

Gacanin et al. demonstrated multifunctional protein-DNA hybrid hydrogel consisting of human serum albumin (HAS-PEG) conjugated to rationally ssDNA sequence linker to deliver recombinant Rho-inhibiting C3 toxin (C2IN-C3lim-G205C) [77]. C2IN-C3lim-G205C is able to selectively reduce osteoclast migration in vitro without affecting the viability, activity, and proliferation of bone-forming osteoblasts. Therefore, it is considered inhibition of the osteoclast’s formation for the local delivery in osteoporosis. This hybrid hydrogel has several advantages, such as self-healing property with favorable injectability, rapid gelation under physiological conditions without any chemical crosslinker and toxic groups, as well as high capability of loading and controlled release through DNA-cleaving enzyme (DNase). Besides, the result showed a significant decrease (≈96%) in the expression of essential osteoclast markers and the resorption activity of osteoclast cells [77]. Electrostatic attractions are considered as the most common mechanism to form self-healing hydrogels for bone tissue engineering. Silk fibroin (SF) showed several benefits for bone-related applications such as strong mechanical properties, biocompatibility with FDA approval, tunable biodegradability, and minimal/nonimmunogenicity [14]. For example, Shi et al. studied dynamic SF-based hyaluronic acid (HA) hydrogel modified with bisphosphonate (BP) groups to be able to bind reversibly to Ca^2+^ ions that were coated onto silk microfibers (mSF) [78]. Although HA is a critical component of the ECM in angiogenesis, wound repairing, matrix organizations, morphogenesis, and cell signaling, one of the severe issues of HA is its high degradation rate in vivo. Consequently, conjugation with synthetic materials could enhance the strength, toughness, durability of HA. Silk-based hydrogels exhibited low mechanical properties, which can be improved by incorporating a UV cross-linkable compound [78].

HA–BP–acrylamide (Am–HA–BP) prepolymer leads to more stability and an increment of almost 15 times in storage modulus. Am–HA–BP hydrogel indicated osteogenic properties in vitro and a considerable increase in bone regeneration. In addition, the formation rate of bone increased to 220% compared to untreated groups in vivo. Biomineralization silk fibroin (SF) through immersing to calcium phosphate with interaction and in the next step, the natural polymer (HA) modified with bisphosphonate (BP) as binders to chelate Ca^2+^ ions prepared double networks that were fully reversible and dynamic (Figure 6A). The autonomous self-healing property was confirmed by macroscopic observation (Figure 6B). Human mesenchymal stem cells (h-MSCs) were seeded on the surface both of Am-HA-BP+mSF and Am-HA-BP·CaP@mSF hydrogels for 14 days. Cells viability was determined by fluorescence stained cells with Phalloidin Tetramethylrhodamine (TRITC). Their results have revealed mineralized mSF increasing cell adhesion and formation of self-healing silk hydrogel occurred in combination with Am-HA-BP filling irregularly shaped bone cavities without the risk of liquid material leakage (Figure 6C). Expression of osteocalcin (OC) as late marker osteogenesis and Col I as an early marker on the DC Am-HA-BP·CaP@ was decreased, while the expression of VEGF, which is strongly associated with the vascularization process during bone repair, was increased in comparison with Am-HA-BP+mSF hydrogel (Figure 6C). Am-HA-BP·CaP@mSF hydrogels were implanted into rat cranial critical defects (diameter: 8 mm) to confirm bone regeneration in vivo during the fourth and eighth week. Although new bone was observed in the implant–tissue interface area, there was not any noticeable mineralized tissue in the untreated group (Figure 6D). Meanwhile, HA-BP and acrylate BP (Ac-BP) with magnesium chloride (MgCl_2_) were reported as electrostatic interactions used to improve the shear thinning property, compressibility, and stress relaxation profile of this Am–HA–BP hydrogel. This nanocomposite hydrogel indicated the differentiation ability of hMSCs to osteoblast [78].

Numerous bioactive molecules such as genes, growth factors, and small molecule drugs, which are crucial in molecular signaling, can be delivered by self-healing hydrogels. Zhang et al. proposed a nanocomposite hydrogel based on conjugation methacrylate HA with thiolglycolated pamidronate (thiol–Pam) and self-assembled Pam-Mg nanoparticles (NPs). Therefore, magnesium ions (Mg^2+^), as a critical cofactor for the enzymatic activity of alkaline phosphatase (ALP), could steadily release and promote differentiation hMSCs to osteoblast. The dynamic organization between Mg2+ and pamidronate (Pam) improved desirable injectability and efficient stress relaxation. Moreover, loading synthetic glucocorticoids dexamethasone (Dex) as pro-drug Dex phosphate (DexP) exhibited effectively sustained release drug from hydrogel, which could improve bone regeneration in a rabbit model [65]. Table 3 summarizes the works conducted on the purpose of bone regeneration using self-healing hydrogels.

### 2.3. Cardiac Tissue Engineering

Myocardial infarction (MI) is considered as having one of the highest mortality rates worldwide. Although natural hydrogels are used for myocardial injection therapies, including collagen, chitosan, fifibrin, alginate, HA, and Matrigel, they demonstrate batch-to-batch differences [79]. Several synthetic hydrogels and self-assembling peptides can potentially deliver cells and drugs by injection and formation in situ sol-gel without any bacterial contamination to restore the damaged myocardium. Recently, among the biomaterials utilized for cell delivery, injectable hydrogels with high conductivity have been broadly studied as potential candidates for cell and drug delivery carriers. Self-healable hydrogels can mimic myocardium with minimally invasive effects, which are helpful for regenerating the damaged myocardium [14].

#### Injectable Hydrogels for Cardiac Tissue Engineering 

Synthetic hydrogels have typically formed via chemical or physical crosslinking, self-assembly, thermal switching, photo-induced polymerization, or noncovalent interactions by dynamic nature of the supramolecular that are held together to control sol-gel switching behavior under mild conditions, e.g., UPy [14]. Bastings and coworkers reported that Catheter-based drug delivery approaches are considerably less invasive than surgical implantation [80].

Catheter delivery of myocardial active growth factor by UPy-modified poly (ethylene glycol) (PEG) chains have been proposed for the treatment of MI. The UPy hydrogels illustrated a self-healing property within minutes. Meanwhile, the storage modulus of the 10 wt.% UPy conjugated with alkyl-urea spacers to 10 k gel matches the mechanical stiffness of the natural cardiac tissue. Furthermore, this hydrogel, as the carrier of growth factors, was able to considerably reduce the size of an infarct scar within a porcine model and improved the activation of resident regenerative cells to boost rapid cardiac tissue regeneration. The results showed that UPy with alkyl-urea spacers to 10 k hydrogel release the growth factors during seven days through an initial 41%, and then a sustained release until 97% at the end [80]. Notably, double network hydrogel containing covalent and noncovalent interactions could improve mechanical strength. For instance, guest–host interactions CD and ferrocene (Fc) have responded to electrical stimulation as well as a photo-switchable crosslinker CD and trans azobenzene, which have altered from the gel phase to the sol phase with light exposer into the infarcted region of cardiac tissue of a sheep model [13,81]. Furthermore, injectable hyaluronic acid (HA) hydrogel with guest–host interactions of CD as host and adamantane (AD) as a guest for the local and sustained delivery of miR-302 mimics the MI mouse heart to promote mammalian cardiac regeneration during two weeks [82]. For self-assemble hydrogels from guest (adamantine, Ad) and host (β-cyclodextrin, CD), following injection, a secondary covalent crosslinking is used in situ through Michael-acceptor reactivity and catalytic conditions in order to stabilize the network pendant groups. This mechanism facilitates shear-thinning delivery with high retention at the MI pig model [83].

Recently, nanomaterial-based hydrogels have been highly noticed from a biomedical point of view to enhance the thermal and electrical conductivity of hydrogels [84]. For instance, the incorporation of carbon nanotubes (CNT) into the photocrosslinkable gelatin methacryloyl (GelMA) hydrogel leads to a stable beating rate and three times greater and more homogeneous F-actin fibers than the pristine GelMA. The result of Shin SR et al.’s study revealed CNT-GELMA hydrogel has an integration ability and improves cell to cell communication among seeding neonatal rat cardiomyocytes on hydrogel [84]. In addition, reversible Schiff base reaction between aldehydeoxidized alginate (ALG-CHO) and amine gelatin groups are able to control H_2_S-gas releasing to aim at the complex symptoms of MI such as increasing vascularization and mechanical performance to adjust to the dynamic cardiovascular condition, tightly integrated with the myocardial tissue [85].

Wu et al. proposed the combination of conductive cardiac patch and injectable self-healing hydrogels consisted of a mixture patch of gelatin-dopamine (GelDA) with ionic coordination conjugated to dopamine-modified polypyrrole (DA-PPy). Schiff base reaction between oxidized sodium hyaluronic acid (HA-CHO) and hydrazide hyaluronic acid (HHA) leads to improving mechanical support, and this hydrogel can enhance the adhesive property and promote angiogenesis MI. The result of the combination internal (hydrogel injection)−external (patch) therapy showed improving storage modulus, conductivity, and gelation time of the hydrogel with outstanding effect in enhancing the cardiac function after the occurrence of MI compared to single-hydrogel systems (Figure 7) [86].

Dong et al. developed a self-healing, electroactive, and biocompatible hydrogel based on an chitosan-graft-aniline tetramer (CS-AT) polymer [79]. The dynamic covalent Schiff-base linkage among chitosan amine groups and PEG-DA benzaldehyde groups acts as an excellent vehicle for cell therapy within cardiac tissues. CS-AT hydrogel showed near conductivity to native cardiac tissue, and encapsulation of myoblasts (C2C12) and cardiac cells (H9c2) revealed a linear-like release profile controlled by the density and type of cells and rapid degradation profile over the span of 45 days without inducing a considerable inflammatory reaction. The proliferation of C2C12 cells was boosted (see Figure 8A–C). As revealed in Figure 8D, the number of cells significantly increased on days 2 and 3 compared to day 1. After injection of hydrogel containing C2C12 cells (1 × 10^6^ cells mL^−1^) via 22 gauge needles (Figure 8E), cells maintained good viability and morphology with confocal microscopy (Figure 8F–H). Since repairing impaired cardiac tissue with more than one kind of cells is desirable, ADMSCs as stem cells (red) and C2C12 cells (green) could inject together and fusion (Figure 8I–L) [79].

Finally, the self-healing hydrogel formation through host–guest interactions of adamantane and beta-cyclodextrin (CD) modified HA were studied to encapsulate endothelial progenitor cells for enhancing neovascularization in the ischemic rat model [83]. According to the result of this study, the stiffness and retention capacity of the hydrogel increased compared to the untreated hydrogel [83]. Furthermore, HA hydrogels modified with CD were developed for siRNA against MMP2 (siMMP2) delivery for the treatment of infarcted myocardium. This structure contains modified HA with CD for host–guest interaction with cholesterol from siRNA so that siRNA could retain for more than two weeks. Moreover, hydrazone bonds have been formed between aldehyde from aldehyde-modified HA (ALD-HA) and hydrazide from hydrazide-modified HA macromer (HA-MMP-HYD), and eroded in response to MMP activity to sustain release of siRNA [87].

Recently, stem cell-derived secretome has been studied as a potential approach to conventional stem cell therapy for regeneration of the myocardial. For instance, nanocomposite hydrogel consisted of Laponite as nanoclay disk-shaped nanoparticles can adjust the release of secretum-loaded hydrogel via electrostatic interactions with hydrogel [88]. In other words, a highly cross-linked gel with Laponite was able to release secretum secreted by human adipose-derived stem cells (hASCs) for a long time to improve angiogenesis and cardioprotection in vitro and in vivo [88]. Table 4 shows some of the studies about heart tissue regeneration using self-healing hydrogels.

### 2.4. Injectable Hydrogels Used for Neural System Applications 

The mammalian central nervous system (CNS) has a weak capability to regenerate neurons or axons after damages [89]. Although stem cell therapy has shown benefits for repairing CNS damages, most of the cell’s loss at the targeted sites limits transplantation of the stem cells. Hydrogels improve cell survival via carrying stem cells, especially self-healing hydrogels with high stability in situ gelations. Electroconductive self-healing hydrogels could regulate migration, differentiation, metabolism, adhesion and, a proliferation of electrically excitable cells, which is critical for neural tissue engineering [90]. For example, nanocomposite self-healing hydrogel from *N*-carboxyethyl chitosan (CEC) with polypyrrole (DCP) nanoparticle (~40 nm) and a unique aldehyde-terminated difunctional polyurethane (DFPU) as linker revealed electroconductive properties in vitro and in vivo. This hydrogel could stimulate proliferation, attachment, and differentiation of neural stem cells (NSCs) [90]. Recently, semi-interpenetrating polymer network (SIPN) hydrogels have been studied for cell encapsulation consisting of linear, branched, and crosslinked polymeric networks [91]. For example, SIPN hydrogels composed of HA were incorporated into the chitosan-based self-healing hydrogel with the appropriate stiffness and good injectability to enhance axonal growth in the zebrafish model’s traumatic brain injury [91]. However, with amyloid nanofibrils with the capability of self-assembly used as biomaterial applications for encapsulation murine neural stem cells and neural-like PC12, weak mechanical properties of the hydrogel limited its application inside the body [91,92]. In another approach, self-healing hydrogel composites made from chitosan–cellulose (CS–CNF) nanofiber dramatically improved their strain sensitivity via the interaction of cellulose nanofibers (CNFs) with the reversible Schiff crosslinking in the CS self-healing hydrogel. Besides, neural stem cells encapsulated in the hydrogel showed significantly better oxygen metabolism and neural differentiation. The results demonstrated that cell viability and cell metabolism in the samples with a low concentration of CNF (CS–CNF1/2) was higher compared to high CNF content samples (CS–CNF3 and CS–CNF4) and pristine CS hydrogel (Figure 9A) [93]. The bioenergetics of NSCs in hydrogels was measured by the OCR values of the cells. Figure 9B has indicated OCR in CS–CNF2 increased compared to cells in the pristine CS hydrogel, which means cells in CS–CNF2 hydrogel produce higher ATP even after the addition of rotenone. Both mitochondrial function and nonmitochondrial respiration were considerably improved in CS–CNF2, whereas it was diminished in CS–CNF4, compared to those in the pristine CS hydrogel (Figure 9C–E) [93].

Some injectable hydrogels with both biological and mechanical properties have been fabricated from collagen type I [94]. Collagen hydrogel crosslinked with 4S-StarPEG (PEG ether tetrasuccinimidyl glutarate) was utilized as a carrier for loading genetically modified rat bone marrow MSCs to over-expressed glial cell line-derived neurotrophic factor (GDNF), which indicated a well-tolerated cell delivery platform [94,95]. Since MSCs delivered to the CNS typically exhibit low survival post-transplantation, this hydrogel improves cell survival and graft integration. However, MSCs cannot survive in scaffold properly and are reduced in volume post gelation in vitro and in vivo for several days [94]. Notably, a self-healing hydrogel composed of dynamic imine bonds between chitosan and oxidized sodium alginate was utilized to carry NSCs into the CNS. It was revealed that it could be to delicately modulate the hydrogel stiffness in the range of 100 to 1000 Pa to exactly fine-tune the differentiation and growth of NSCs in a mice model [96]. This system led to a homogenous cell distribution when the NSC-loaded hydrogels were injected into mice model brains. Due to the fact that electrical properties are considered as vital properties for the engineering of neural tissues, some studies focused on the mechanical and electrical features in microenvironments to help the progress of the neural cells. In this approach, Hou et al. formulated a graphene-based self-healable electroconductive hydrogel to assure the growth of neural-like PC12 cells under nonphysiological conditions [97]. Thus, further investigation is being done on self-healing and electroconductive hydrogels for neural tissue applications [97]. The percentage of transplanted survival cells is limited by acute inflammation after injury or transplantation. Therefore, an intelligent double-layer alginate hydrogel system was designed in which an inner layer modified with alginate MMP and RGD polypeptides to improve neural stem cells (NSCs) adhesion, and an outer layer contains Cripto-1 antibodies to facilitate differentiation of NSCs to dopaminergic neurons for treatment of Parkinson’s disease [98]. In summary, the few covered studies here revealed the feature of injectable self-healable hydrogels to modulate growth, survival, and differentiation of neural cells to target stem cell therapy for the degenerated CNS. It is essential to use the potential of biomaterials for biomedical applications. Table 5 contains a summary of self-healing hydrogel application in neural tissue engineering. 

### 2.5. Injectable Hydrogel Used for Lung-Related Applications 

The lung is a structurally and functionally complex organ [98]. Lung disease is among the leading global causes of death [99] and a health concern that severely affects patients’ quality of life [100]. There are limited treatments for lung diseases, including cystic fibrosis (CF), chronic obstructive pulmonary disease (COPD), and idiopathic pulmonary fibrosis (IPF), and lung transplantation is often the only option for end-stage patients [101]. Acute respiratory distress syndrome (ARDS) is a major cause of failure from lung endothelial and epithelial permeability, leading to hypoxemia, pulmonary oedema, and loss of lung compliance [102,103]. ARDS occurs in above 10% of intensive care unit (ICU) patients worldwide, and some of them receive mechanical ventilation in the ICU [104]. ARDS can also be developed by a pulmonary infection or a trauma that leads to providing pro-inflammatory cytokines, which can improve acute organ dysfunction [105].

Although ARDS is a serious illness with a large rate of incidence, no direct therapies have been developed for it [106]. Currently, the most commonly used strategies are protective mechanical ventilation and fluid-restrictive, which improve oxygen perfusion in the lungs [107]. Meanwhile, pharmacologic treatments such as surfactants, glucocorticoids, antibiotic therapy, antioxidants, and a wide range of other anti-inflammatory treatments have been indicated as completely ineffective [108].

One of the most common problems is a lack of lung tissue for transplantation. In addition, transplant recipients have to use immunosuppressive drugs for the rest of their lives, which can lead to problems. Recently, a new approach called Cell Formation in Lung Tissue Engineering has been proposed. The main components of tissue engineering include (I) a proper biological or artificial 3D scaffold, (II) the source of cells or stem cells, (III) the growth factors necessary for cell differentiation and proliferation, and (IV) bioreactor, which supports a biologically active 3D composite [100,101]. Hydrogels are among the components used for scaffolding (Figure 10) [100].

The use of biological scaffolds retains many of the complexities of combining extracellular matrix (ECM) and biological activity. Hydrogels are quite suitable for making lung tissue cell scaffolds due to their proper structure and mechanical properties similar to the extracellular matrix (ECM) of soft tissue. Collagen and hyaluronic acid are among the compounds used to make hydrogels suitable for soft tissue, which is widely used due to their biocompatibility and biodegradability [109].

The superior generation of hydrogels is the self-assemble type, which are sensitive to temperature, PH, and salt concentration and are formed at physiological body temperature [110]. Collagen I hydrogel matrix and fibrinogen-fibronectin-vitronectin hydrogel (FFVH) are among lung tissue scaffolds. 

The FFVH scaffolds are suggested in end-stages of lung disease, and their results indicate proper adhesion and distribution of cells in the extracellular matrix using this model of hydrogel scaffold in the laboratory. Collagen I hydrogel matrix scaffolds differentiate mesenchymal stem cells into epithelial and endothelial lineages more efficiently and also provide better cell preservation. ECM compatible natural hydrogels such as collagen I and Matrigel gels as 3D matrices are suitable to differentiate lung cells and lung tissue morphogenesis due to their mechanical and biochemical properties [100].

The use of ECM hydrogel as a cell scaffold for lung cells has shown favorable results. In a study by Link et al., the use of the appropriate concentration of genepin has led to more similarity between the hydrogel made and lung tissue [110]. Dunphy et al. also conducted a study involving collagen-elastin structures designed to match the properties of an alveolar wall. The study showed how elastin affects the stiffness of collagen hydrogels, and the inclusion of pulmonary fibroblasts in these structures shows similar results to an alveolar wall [99]. Pouliot et al.’s study confirmed that hydrogels obtained from de-cell lungs are very promising for extracellular modeling of the lung in vitro and the development of clinical therapies [101].

The use of cellulosic scaffolds for tissue-specific ECM isolation (dECM) for tissue engineering as modern clinical therapies and the improvement of in vitro ECM study models is increasing. Most ECM hydrogels are produced using an overall strategy in which the decellularized tissue is lyophilized, ground, and then enzymatically, most often with pepsin, placed in an acidic environment for 24–72 h (hours) to achieve proper solubility. Recently, lung ECM hydrogels have been developed as a model for the study of ESM obtained from diseased and healthy lungs, as a protective treatment for radiation-induced lung damage in vivo, and as bio links for the production of three-dimensional additives. This highlights the potential of ECM lung hydrogels for impact in several areas. ECM hydrogels are easily regenerated by immune cells, which is one of hydrogels’ advantages [101].

The combination of natural biological materials alginate and gelatin with a carbodiimide cross linker is being considered as a new concept for tissue engineering. High molecular weight polymer alginate is biocompatible and nontoxic. Gelatin is a natural water-soluble polymer and is one of the most widely used materials for tissue engineering applications due to its biodegradability and biocompatibility. The Shulimzon study confirms the effectiveness of three-dimensional hydrogel scaffolds with gelatin-alginate origin in lung tissue regeneration [111]. Many researchers and physicians believe that the use of scaffolding improves tissue regeneration and can be useful in lung diseases such as COPD. Moreover, mesoporous silica scaffold and poly(lactide-co-glycolide) matrices were fabricated to modulate the immune cell function for tumor vaccines in animal models with hopeful results and may apply as a platform for the design of vaccines against SARS-CoV-2 [112]. Recently, the polymer-nanoparticle (PNP) hydrogel system has been able to sustain antigen release with strong antibody responses to adjuvanted subunit antigens considered as injectable vaccines. Gale et al. used the receptor-binding domain (RBD) of the SARS-CoV-2 spike protein (10 µg) as antigen and FDA-approved adjuvants CpG and Alum, which is loaded in injectable self-healing hydrogel from hydroxypropylmethylcellulose (HPMC-C12) combination with poly(ethylene glycol)-*b*-poly(lactic acid) (PEG-PLA). Their results indicated slow release of adjuvant and antigen during 18 days leads to the highest titers neutralizing antibody [113]. The study of Wu et al. shows that HTCC temperature-sensitive hydrogels are highly promising for delivering H5N1 influenza split antigens (A/aNhui/1/2005(H5N1)) with a concentration of 150 μg/L and vaccination against influenza through internasal administration in female Balb/c mice [114].

Since the respiratory system is exposed to the external environment and microorganisms (such as bacteria, viruses, and fungi) in the air, it causes respiratory infections and diseases around the world [115]. Nanogels have been suggested to tackle this problem by loading antibiotic agents in hydrogel to protect the antimicrobial agents from deactivation and diminish the adverse effects by decreasing the drug exposure to the rest of the body. Nanogels are formed by physical and chemical crosslinking of polymers, making them more promising for biomedical applications [116]. Chen et al. proposed an injectable self-healing PEG thiolated hydrogel for sustained release clarithromycin (CAM) and budesonide (BUD) in rabbit models of acute bacterial rhinosinusitis (ABRS) with *Staphylococcus Pneumonia*, which during two weeks indicated a therapeutic effect. The injectable self-healing capability PEG hydrogel depends on the high affinity, and reversible binding between sulfhydryl groups and silver ions give rise to control release drugs without adverse side-effects as well as inhibition inflammatory responses [117]. Additionally, injectable smart hydrogels based on poly (ε-caprolactone-*co*-lactide)ester-functionalized hyaluronic acid (HA-PCLA) for delivery immunomodulatory factor (OVA expressing plasmid, pOVA) could induce production antibody and effective inhibition of human lung carcinoma in vivo [118]. Table 6 summarizes selected studies of self-healing hydrogels for lung tissue engineering.

### 2.6. Injectable Hydrogel Used for Wound Healing Applications

Physical or thermal damage leads to chronic wounds, especially in people who suffer from diabetes. Hydrogels have benefits for treatment and reducing inflammation because of excellent permeability, excellent biocompatibility, and providing a wet environment for wound repair. In addition, antioxidant hydrogels could decrease the levels of reactive oxygen species (ROS) to prevent oxidative stress and subsequently repair the wound [119]. Li et al. suggested that sodium alginate/ZnO hydrogel beads have the capability for sustained release of curcumin. These hydrogel beads indicated pH sensitivity and controlled-release ability curcumin with high antioxidant activity [120]. Liang et al. showed that gelatin-grafted-dopamine (GT-DA) and polydopamine-coated carbon nanotubes (CNT-PDA) have antibacterial, adhesive, antioxidant, and conductive ability. GT-DA/CS/CNT hydrogel could release antibiotic doxycycline for treatment of full-thickness defect wounds [121]. Moreover, N-deacetylated derivative of chitosan with functional groups could bioconjugate with oxidized chondroitin sulfate (OCS) and make injectable, self-healing, and antibacterial hydrogel newtwork for drug delivery without any chemical crosslinking. Li and coworkers showed that N,O-carboxymethyl chitosan/oxidized chondroitin sulfate (N,O-CMC/OCS) hydrogel have a long gelation time (133 s), inherent antibacterial and stable performances properties for fibroblast cells, and that endothelial cell delivery damages skin [122]. Recently, Kaolin has been utilized as a blood clotting stimulation via its negative charge contact to the factor XII and platelets. Tamer et al. indicated hemostatic and antibacterial properties of polyvinyl alcohol (PVA/Kaolin) hydrogel. Kaolin could improve the swelling capacities of hydrogel as well as pores sizes of the fabricated hydrogels, which lead to the absorption of wound exudates. Antibacterial property of this hydrogel could be boost by loading penicillin streptomycin (Pen-Strep) for prevention of skin infections [123]. Although numerous hydrogels with inherent antibacterial capability or loading anti-bacterial drugs have been used, being drug-resistant limits it from being widely used and is considered a critical issue. Liang and coworkers proposed graphene oxide (GO) loading in gelatin methacrylate (GM) and glycidyl methacrylate functionalized quaternized chitosan (QCSG) to overcome drug resistance in damaged skin with Staphylococcus aureus (MRSA) infected mouse. Their results indicated GO with negative charge on the surface and good photothermal features, and antibacterial properties could repair damaged skin. Besides, methacrylate groups could increase mechanical properties of gelatin [124]. Table 7 summarizes selected studies of self-healing hydrogels for wound healing.

## 3. Conclusions

This review has provided a glimpse of the numerous applications of self-healing hydrogels as biomaterials, which are highly desirable because of useful advantages such as recovering their shape and mechanical properties after damage, moldability, and smooth injectability applications. The aim of current strategies is to design and fabricate applications that can mimic the in vivo conditions of different tumor types and restore damaged tissues and organs. Various research sought to create sensitive hydrogels to control the delivery system of anti-cancer drugs at the tumoral site. However, multiple features are required to accurately lead to this aim, including immunological response, the reaction time, degradation rates, surface hybridization, and inflammation reactions.

The studies were outlined on supramolecular interactions of hydrogen to enable tissue ingrowth and cell migration, improving the hydrogels’ capability to thrive in physiological environments and anti-cancer drug delivery. Although these hydrogels promote proliferation, differentiation, and cell spreading, they do not have enough research in the clinical phase. Since the amassed demands for completely imitating the structure for cell development and growth and sustain drug release, the applications of self-healing hydrogels have been increased. The scope of this review is interdisciplinary among chemistry, medicine, physics, nanoscience, biology, and mechanical engineering, which altogether can address some of the current issues.

## 4. Future Prospective

Desirable scaffold structure and biological function have improved through concurrent advancements in vascularization and immunomodulation. Markedly, the designee of biocompatible polyethylene glycol hydrogel base on CRISP system with single-stranded DNA and endonuclease cas12 could develop sustained release drugs, nanoparticle delivery [125]. It is expected that in the near future, conventional hydrogels will be replaced with intelligent self-healing hydrogels to solve the problem of UV-light in crosslinking, which is usually unable to diffuse the deeper parts of the body.

## Figures and Tables

**Figure 1 polymers-13-02680-f001:**
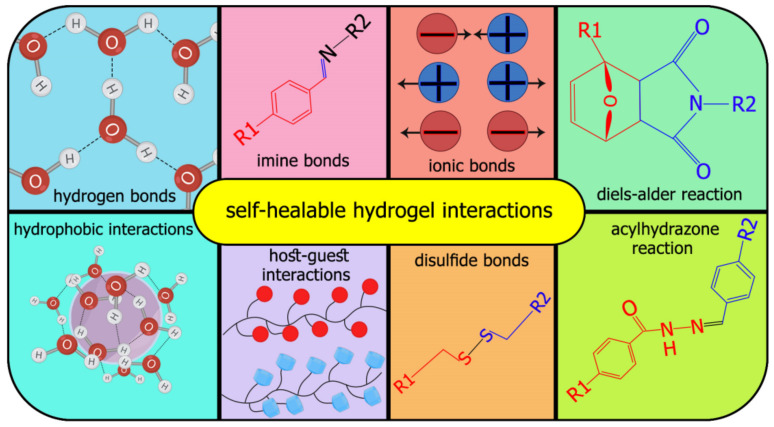
The numerous chemical and physical interactions behind self-healable hydrogels.

**Figure 2 polymers-13-02680-f002:**
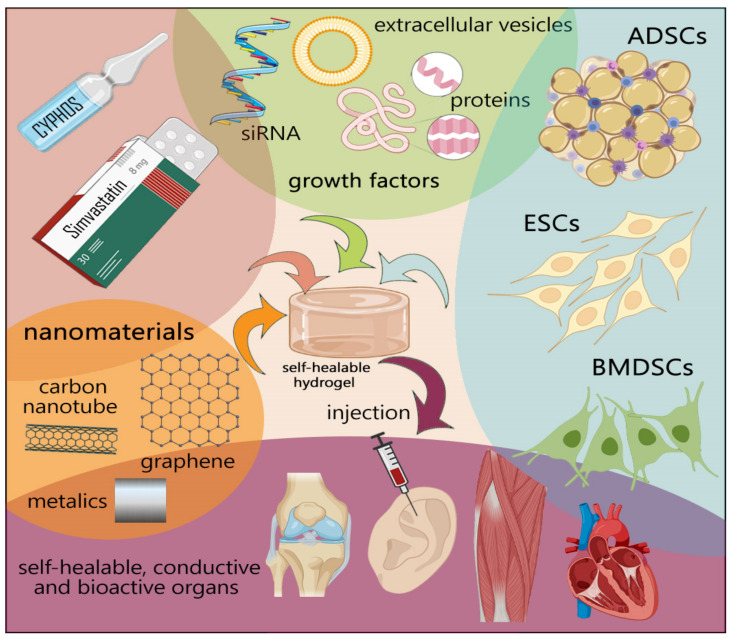
Broad overview on tissue engineering. Tissue engineering contributes to cell therapies, gene editing, genetic manipulation, and nanomedicine.

**Figure 3 polymers-13-02680-f003:**
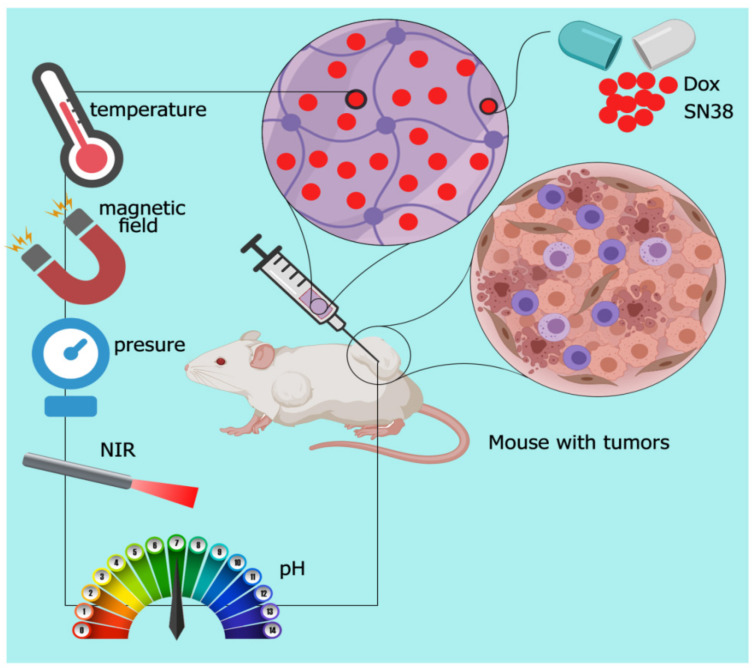
Schematic of external-responsive hydrogels. Combination therapy through sustained released anti-cancer drug from smart hydrogel, which changes formation in response to the external stimulations.

**Figure 4 polymers-13-02680-f004:**
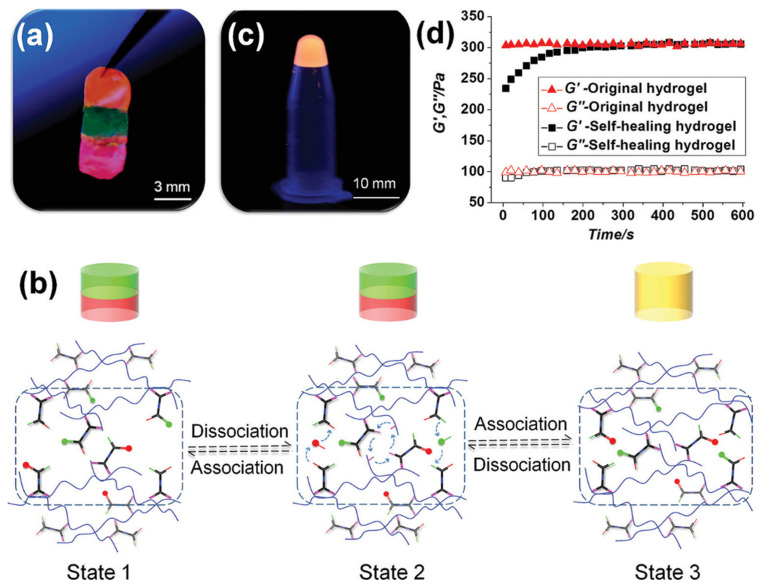
Formation polypeptide−DNA hydrogel with self-healing property, (**a**) the ability of adherence freshly prepared hydrogels (4 wt.%), from bottom to top are 5(6)-ROX modified, 5(6)-FAM modified, and 5(6)-FAM/5(6)-ROX modified polypeptide-DNA hydrogels, respectively, (**b**) mechanism of self-healing through DNA chain exchange, (**c**) three hydrogels treated with dyes, (**d**) mechanical performance of polypeptide−DNA hydrogel. Reused with permission from [71] Copyright © 2021 Small.

**Figure 5 polymers-13-02680-f005:**
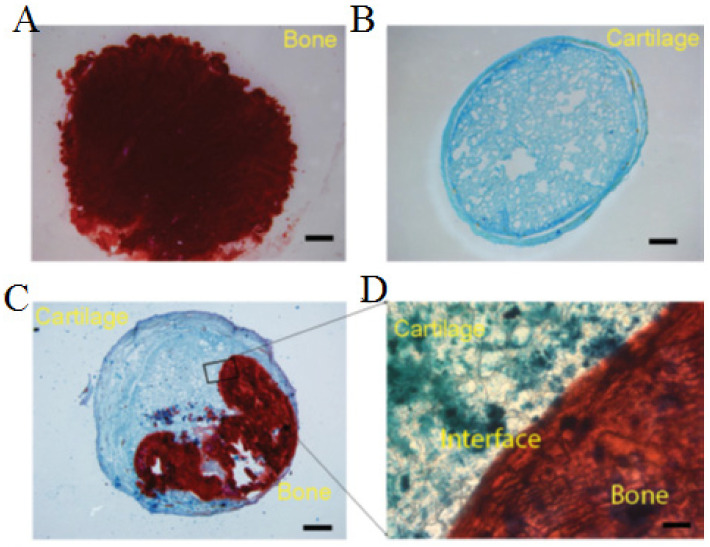
Formation self-healing hydrogel of Dextran-Ureidopyrimidinone (Dex-Upy) polymer. The bone–cartilage interface to the formation of osteochondral constructs in vivo. (**A**) Only group completely positive for Alizarin red staining to detect bone mineralization. (**B**) Sectioning cartilage alone completely positive for Alcian blue staining to detect cartilage tissue. (**C**) The bone–cartilage interface to the formation of osteochondral constructs in vivo. (**D**) Magnify the bone–cartilage interface. Reused with permission from [39] Copyright © 2021 Advanced Healthcare Materials.

**Figure 6 polymers-13-02680-f006:**
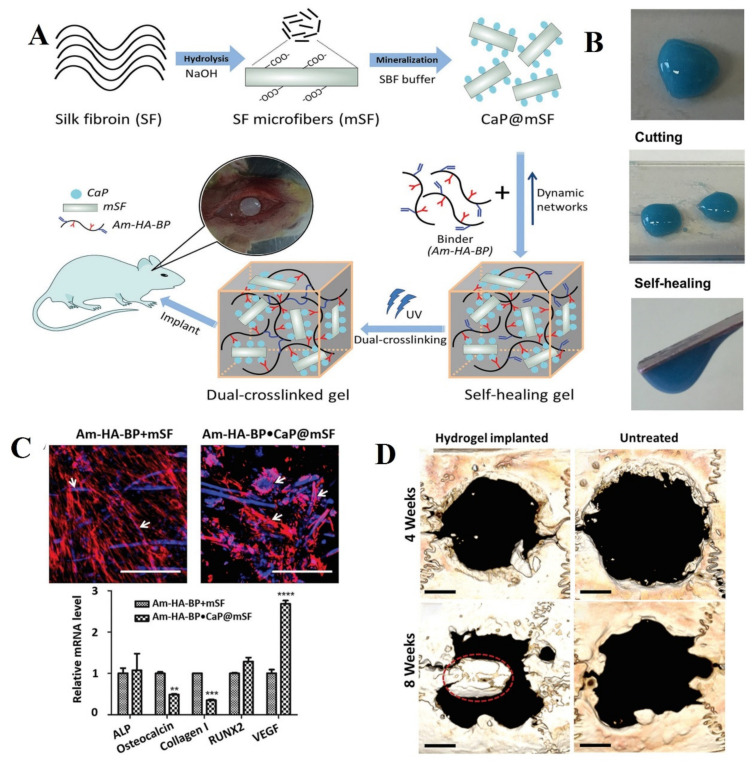
Schematic of silk self-healing hydrogel for bone tissue. (**A**) The design of the self-healing hydrogel formation. (**B**) Illustration of the ability of self-healing hydrogel after cutting into pieces. (**C**) The hMSCs expanding within the hydrogels were evaluated by staining cell cytoskeleton with phalloidin (red) and nucleus with Hoechst (blue). The relative osteogenesis gene expression was assessed from hMSCs on the hydrogel. (**D**) The improvement of bone formation in a rat cranial defect site after treatment with self-healable hydrogel. Reused permission with from [78] Copyright © 2021 Advanced Functional Materials.

**Figure 7 polymers-13-02680-f007:**
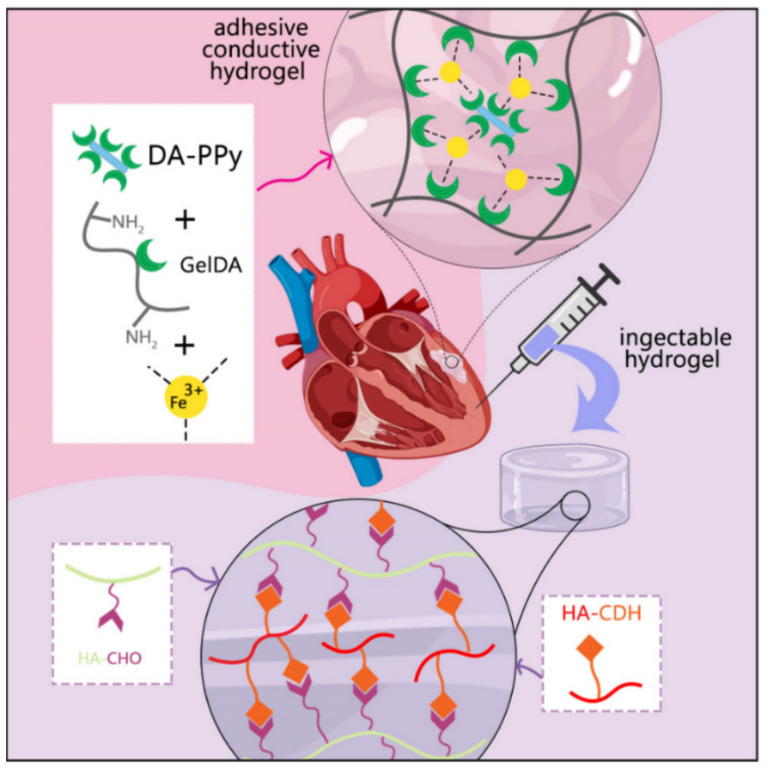
Effect of combination therapy via adhesive hydrogel patch and injectable hydrogel for therapy of MI. The highest quantitative microvascular density and small arterioles were shown in combination hydrogel via the expression of the Willebrand factor (vWF). Adapted with permission [86].

**Figure 8 polymers-13-02680-f008:**
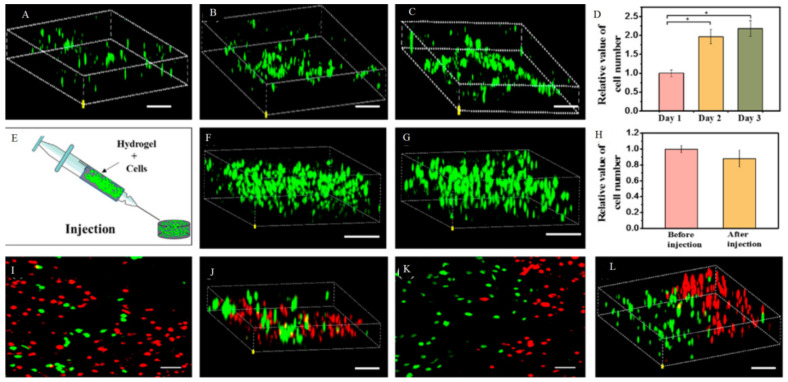
Confocal image of cell proliferation of the C2C12 myoblasts encapsulated in the CS-AT10 hydrogels. (**A**) Visualization of cells proliferation during 1 (**B**) 2 and 3 days (**C**). Proliferation curve of the C2C12 myoblasts encapsulated in the hydrogel (**D**). Injection self-healing hydrogels with encapsulation cells (**E**) and visualization cells before (**F**) and after injection (**G**). The diagram indicated conductive hydrogel has the highest live cell numbers in the hydrogel after injection (**H**). Visualization two type of cells encapsulated injection hydrogel (**I**–**L**). Reused permission with from [79] Copyright © 2021 American Chemical Society.

**Figure 9 polymers-13-02680-f009:**
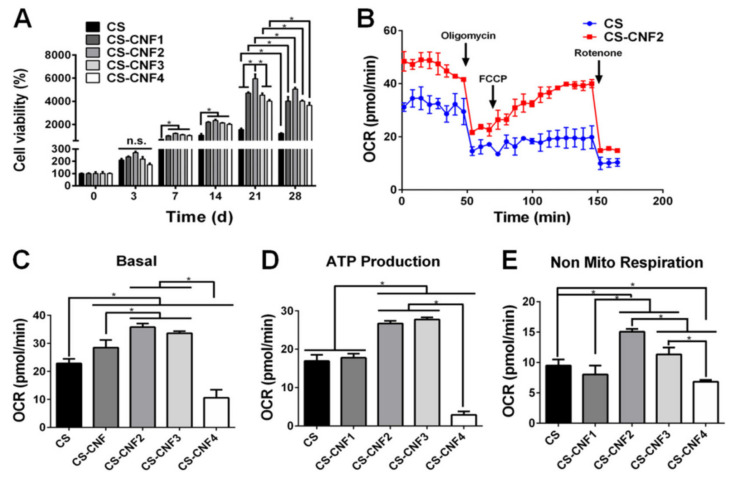
The cell growth ability and oxygen metabolism of neural stem cells (NSCs) in different CS-CNF nanofiber hydrogels contents of CNFs (0.06, 0.09, 0.12, and 0.15 wt%) (**A**) The measuring cell viability and proliferation measurement by the CCK-8 assay. (**B**) The oxygen metabolism of the cells inside hydrogels was evaluated via the oxygen consumption rate (OCR) marker and oligomycin, FCCP, and rotenone used as an inhibitor. (**C**–**E**) Evaluation of the OCR parameters for NSCs in different self-healing hydrogels indicated Basal OCR values, ATP production, and nonmitochondrial oxygen consumption increased in CS-CNF in computation CS and highest concentration CNF4(0.15 wt%). Reused permission with from [93] Copyright © 2021 NPG Asia Materials.

**Figure 10 polymers-13-02680-f010:**
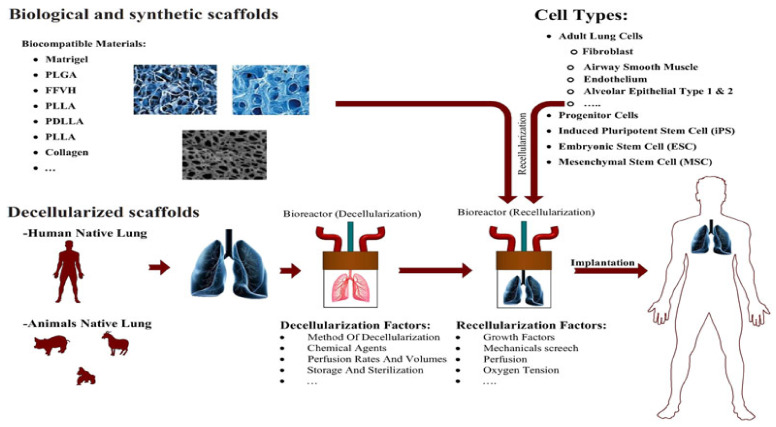
Design of effect of natural scaffolds for lung tissue via the best method of decellularization with different biocompatible materials. Different natural and synthetic scaffolds have been applied for lung tissue engineering with different kinds of cells. Reused permission with from [100] Copyright © 2021 Journal of Cellular Physiology.

**Table 1 polymers-13-02680-t001:** A list of the most common commercially available natural self-healing hydrogels.

Bioimplant	Scaffold Fabrication	Materials	Application	Ref.
SUPPRELIN LA	Hydrogel fibers	2-hydroxyethyl methacrylate, 2-hydroxypropyl methacrylate, and 50 mg of histrelin	The treatment of CPP during 12 months	[6]
PHEMA	Hydrogel fibers	PHEMA/beta-CD hydrogels	Ocular drug admission	[8]
HS-TENG	Freeze drying	PVA/PDAP/MWCNT	Wearable application	[10]
TC-Gel	Hydrogel nanoparticles	TA@CNCs/PANI	Sensor for electronic skin devices	[11]
PDMS/PEDOT:PSS/PAA	3D Printing	polyacrylamide and poly(3,4-ethylenedioxythiophene) polystyrene sulfonate hydrogel	Neural attachment	[12]

**Table 2 polymers-13-02680-t002:** Examples of self-healing hydrogels for cancer drug delivery.

Self-Healing Mechanisms	Materials	Application	Advantages	Disadvantages	Ref.
Imine bond	CEC/PEGDA20 hydrogel	Kinetic release drug Dox (2.0 mg Dox/mL hydrogel)	Suitable gelation time and mechanical strength	-	[35]
Thiol bond	CSSH/Cur-Lip	Kinetic release drug Curcomin (200 μM)	Good cytocompatibility	-	[41]
Thiol bond	BSA hydrogel	Sustain release drug DOX 100 µg/mL of Dox in 200 µL hydrogel	Biocompatibility, biodegradable, viscoelasticity	Low mechanical strength	[43]
Imine bond	Chitosan-PEG	Sustain release TOX(0.37 g TOX) was dissolved in 1800 μL	Biocompatibility, biodegradable	Low mechanical strength	[44]
Imine bond	PEI/PDMAEMA/PDA	Sustain release DOX(1 mg/mL)	Biocompatibility, biodegradable, thermosensitive	-	[48]
Imine bond	CSMA/BPEI-GO	Loading efficiency 60% (DOX)	Biocompatibility, biodegradable, thermosensitive	-	[49]
Ionic interaction	(CAT-Gel)/Fe	DOX (5 mg/kg) and DTX (25 mg/kg) entrapped in CAT-Gel	Biocompatible, non-hemolytic nature	-	[51]
Ionic interaction	MDP	CDN (20 μg/30 μL)	Biocompatible, sustaine release drug	Low mechanical strength	[54]
Enzymatic reaction	Fibrin gel	aCD47-Cy5.5 (50 μ per mouse)	Biocompatible, in situ formation therapeutic gel at the tumor resection site, inducing systemic immunological responses	-	[55]
Imine bond	mPEG-b-PELG	Co-delivery IL-15 1 μg/mL and CDDP 10 μg/mL in vitro and hydrogel co-loaded with IL-15 (0.5 mg·kg^−1^) and CDDP (1 mg·kg^−1^) (Gel + IL-15/CDDP) in vivo	Biocompatibility, biodegradable, In-situ gelation process at body temperature	-	[56]

**Table 3 polymers-13-02680-t003:** Examples of self-healing hydrogels for bone regeneration.

Self-Healing Mechanisms	Materials	Application(s)	Ref.
Imine bond	Nanocomposite oxidized alginate (OA) With DNA nucleotide	Inducing osteogenic differentiation and migration of human adipose-derived stem cells	[72]
Hydrogen bonds	Dextran polymer polysaccharide (DEX) with multiple pendant UPy	Bone regeneration in a nude mouse model	[39]
Electrostatic interactions	Polypeptide backbone derived from human serum albumin	Treatment osteoporosis	[75]
Electrostatic interactions	SF-HA	Bone regeneration as a carrier of cell and drug delivery in the rate model	[78]
Thiol-ene	HA-Pam-Mg	Bone regeneration as a Dex drug delivery in a rabbit model	[65]

**Table 4 polymers-13-02680-t004:** Examples of self-healing hydrogels for heart regeneration.

Self-Healing Mechanisms	Materials	Application(s)	Ref.
Hydrogen bonds	UPy-10 k gel	Carrier of growth factors (HGF and IGF-1) for MI	[80]
Host-guest	HA-CD/AD	Delivery mi-RNA302 for MI ina mouse infarction model	[82]
Host-guest/Thiol	Adamantane-thiol-HA β-cyclodextrin-methacrylate-HA	Tissue repair in pig model of MI	[83]
Photo crosslink	CNT-GELMA	Enhancing CM and stable beatine rate for MI	[84]
Imine bond	ALG-CHO and amine gelatine hydrogel	Improving neovascularization in MI	[85]
Imine bond	Mixture patch of gelatin-dopamine (GelDA) with dopamine-modified polypyrrole (DA-PPy) in hyaluronic hydrogel	Mechanical support and promoting angiogenesis MI	[86]
Imine bond	CS-AT-PEG-DA	Improving inflammatory reaction MI	[79]
Hydrazone band	ALD-HA/HA-MMP-HYD	Carrier of siRNA (MMP2) for MI in a rat infarction model	[87]

**Table 5 polymers-13-02680-t005:** A summary of self-healing hydrogels used in vitro and in vivo for neural tissue engineering.

Self-Healing Mechanisms	Materials	Application(s)	Ref.
Imine bond	CEC-DFPU/DCP	Neural repair in zebrafish brain injury model	[90]
Imine bond	SIPN	Neural repair in the zebrafish brain model	[91]
Imine bond	CS–CNF	Neural stem cells encapsulation and differentiation	[93]
Imine bond	collagen-4S-StarPEG	Stem cell therapy for the degenerated CNS	[95]
Ionic interaction	calcium alginate gel beads	Neural stem cells encapsulation and differentiation in a mouse model	[98]

**Table 6 polymers-13-02680-t006:** A summary of self-healing hydrogels used in vitro and in vivo for lung tissue.

Self-Healing Mechanisms	Materials	Application(s)	Ref.
Ionic interaction	HPMC-C12/PEG-PLA	COVID-19 vaccine for sustained release antigen RBD	[113]
Ionic interaction	HTCC hydrogel/split antigen	H5N1 influenza vaccine for sustained release H5N1 antigen	[114]
Ionic interaction	CAM@Hydrogel/silver	Antibacterial and anti-inflammatory properties	[117]
Imine bond	HA-PCLA	Delivery of pOVA to inhibit human lung carcinoma	[118]

**Table 7 polymers-13-02680-t007:** A summary of self-healing hydrogels used in vitro and in vivo for wound healing.

Self-Healing Mechanisms	Materials	Application(s)	Ref.
Ionic interaction	Sodium alginate/ZnO	Atioxidant wound healing	[120]
Ionic interaction	GT-DA/CS/CNT/Doxy	Infected wounnds healing	[121]
Imine bond	N,O-CMC/OCS	Infected wounnds healing	[122]
Ionic interaction	PVA/Kaolin	Infected wounnds healing	[123]
Ionic interaction	QCSG/GM/GO	Infected wounnds healing	[124]

## Data Availability

Data is contained within the article.

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
