# Peer review of "Multifunctional and Self-Healable Intelligent Hydrogels for Cancer Drug Delivery and Promoting Tissue Regeneration In Vivo"

_polymers, 2021, doi:10.3390/polym13162680_

Round 1

Reviewer 1 Report

With regard to the review article polymers-1330941 entitled "Multifunctional and self-healable intelligent hydrogels for cancer drug delivery and promoting tissue regeneration in vivo", the hydrogels is very interesting topic to discuss, particularly self-healable hydrogels. However, some comments appended below should be considered and carefully addressed before publishing in the Polymers Journal:

  1. The advantages and pitfalls of hydrogels in various biomedical applications should be elaborated.
  2. Hydrogels designed for the wound healing process should be discussed, including hemostatic, antibacterial and antioxidant hydrogels since major formulated hydrogels had outstanding performances on account of the structure and configuration of the hydrogel matrices.
  3. The authors are encouraged to highlight some of the important biomaterials for hydrogels preparation and implementation in the respective applications. For instance, one table includes some of biomaterials, preparation, boosting by bio-agents, applications and the respective references.       
  4. Many articles in relation to the implementation of hydrogels in various biomedical applications have been recently published. The authors should focus on articles published in the last two years to postulate substantial and contemporary information.  For instance, please check these articles provided by various authors to support your review with figures or discussion (https://doi.org/10.1038/s41598-021-82963-1; https://doi.org/10.1016/j.msec.2020.111324;   https://dx.doi.org/10.1021/acs.biomac.9b01732).

Author Response

June 9, 2021

Prof. Alexander Böker,

Editor-in-Chief, Journal of Polymers,

Dear Professor Böker,

Please find attached our revised manuscript entitled “Multifunctional and self-healable intelligent hydrogels for cancer drug delivery and promoting tissue regeneration in-vivo ".We are very pleased to have this opportunity to resubmit our manuscript according to all the suggestions of the reviewers. Below we have provided a point by point response to reviewers’ comments and yellow and green highlighted text indicates where changes have been made in the revised manuscript.

Reviewer: 1

Comments to the Author

 1-The advantages and pitfalls of hydrogels in various biomedical applications should be elaborated.

According to the esteemed reviewer’s comment, the manuscript was checked for benefits and disadvantages.

Although hydrogels have many useful properties, traditional hydrogels are often not ideal for tissue engineering applications due to their irreversibility of covalent crosslinks, leading to easily broken during normal operation (6, 7). This phenomenon is like the same healing process of natural organisms via the reversibility of dynamic covalent crosslinks without an external stimulus, e.g., low pH, enzymatic environment, electric fields, temperatures, and UV light (18). These hydrogels mimic the 3D ECM to regenerate after minor injuries similar to native tissues. In other words, self-healing capability refers to a gel's ability to recover to its original shape via noncovalent interactions, dynamic covalent, and physical bonds.

2-Hydrogels designed for the wound healing process should be discussed, including hemostatic, antibacterial and antioxidant hydrogels since major formulated hydrogels had outstanding performances on account of the structure and configuration of the hydrogel matrices.

The authors would like to appreciate the esteemed reviewer for his/her comment. 2.6 Injectable hydrogel used for wound healing applications part was written according to the esteemed reviewer’s comment.

3-The authors are encouraged to highlight some of the important biomaterials for hydrogels preparation and implementation in the respective applications. For instance, one table includes some of biomaterials, preparation, boosting by bio-agents, applications and the respective references.

  The authors would like to appreciate the esteemed reviewer for his/her comment. Tabl1 was written according to the esteemed reviewer’s comment  

  The most common water soluble polymers include poly(vinylpyrrolidone), poly(acrylic acid), poly(vinyl alcohol), poly(ethylene glycol), polyacrylamide andsome polysaccharides. Many hydrogels have been patented for drug delivery like SUPPRELIN LA which is utilized for the treatment of children with central precocious puberty (CPP) trough sustained release histrelin acetate (6).  Although hydrogels have many useful properties, traditional hydrogels are often not ideal for tissue engineering applications due to their irreversibility of covalent crosslinks, leading to easily broken during normal operation (6, 7). Regarding to conventional hydrogel-based contact lenses reveal fairly low drug loading ability and often lead to burst release drugs for ocular administration, some factors such as control hydrophilic/hydrophobic copolymer ratio, impregnating drug-containing colloidal structures, incorporating ligand-including hydrogels and developing multilayered hydrogels improve release drugs. Incorporation hydrogel poly-2-hydroxyethylmethacrylate (PHEMA) with b-cyclodextrin (beta-CD) hydrogel in contact lenses exhibited longer residence time of H1-antihistamines compared to conventional PHEMA hydrogel via enhancing swelling ratio and tensile strength (8). Furthermore, natural hydrogels for biosensor applications could overcome the limitation of traditional polymers due to biocompatibility, their inherent renewable, nontoxic features, and biodegradability (9). Soft hydrogel based self-healing triboelectric nanogenerator (HS-TENG) consist of photothermic-active polydopamine nanoparticle and multiwalled carbon nanotube (MWCNT) allows inside poly(vinyl alcohol)/agarose hydrogel to enhancing mechanical and electrical property for wearable applications. Feature of self-healing is improved by NIR light and the exposure of water spray (10). TC-Gel sensor as electronic skin devices was prepared from polyaniline (PANI) network and covalently cross-linked poly (acrylic acid) (PAA) network which tannic acid coated cellulose nanocrystals (TA@CNCs) was integrated inside network as reinforcing physical cross-linkers to obtain superior mechanical and electrical property. The dynamic hydrogen bonds TA@CNCs leads to electrostatic interactions in the hydrogel networks to easily reformed at room temperature after broken and providing excellent autonomous self-healing capability high stability contact on human skin (11). Meanwhile, coating implant with conductive hydrogels leads to better interfaces with neural tissues. For example, polyacrylamide and poly(3,4-ethylenedioxythiophene) polystyrene sulfonate have revealed the excellent adhesion to implant (polydimethylsiloxane) which is commonly used as a substrate in soft neural interfaces (12). Numerous different applications natural self-healing hydrogels have been organized Table 1 which is available in marketing.

Bioimplant

Scaffold Fabrication

Materials

Application

Ref

SUPPRELIN LA

Hydrogel fibers

2-hydroxyethyl methacrylate, 2-hydroxypropyl methacrylate, and 50 mg of histrelin

the treatment of CPP during 12 months

(6)

PHEMA

Hydrogel fibers

PHEMA/beta-CD hydrogels

Ocular drug admission

(8)

HS-TENG

Freeze drying

PVA/PDAP/MWCNT

wearable application

(10)

TC-Gel

Hydrogel nanoparticles

TA@CNCs/ PANI

sensor for electronic skin devices

(11)

PDMS/ PEDOT:PSS/PAA

3D Printing

polyacrylamide and poly(3,4-ethylenedioxythiophene) polystyrene sulfonate hydrogel

neural attachment

(12)

4-Many articles in relation to the implementation of hydrogels in various biomedical applications have been recently published. The authors should focus on articles published in the last two years to postulate substantial and contemporary information.  For instance, please check these articles provided by various authors to support your review with figures or discussion (https://doi.org/10.1038/s41598-021-82963-1; https://doi.org/10.1016/j.msec.2020.111324;   https://dx.doi.org/10.1021/acs.biomac.9b01732)

To address the reviewer’s comment, 2.6 Injectable hydrogel used for wound healing applications part was written with

Reviewer 2 Report

The reviewed manuscript presents the state-of-the-art progress of multifunctional and self-healable hydrogels for applications in cancer therapy, tissue engineering, and regenerative medicine. The scope of this review is interdisciplinary, covering chemistry, medicine, physics, nanoscience, biology, and mechanical engineering.

The manuscript is interesting and worth publishing with a slight improvement. In line 63 the authors use the term "supermolecular hydrogels" but it seems that "supramolecular hydrogels" is more appropriate. Loh et al. in ref. 14 cited in the peer-reviewed review also use "supramolecular". Anyway, later in the article, e.g. in Conclusions, the authors already use the term supramolecular.

Author Response

Peer Reviewer: 2

The manuscript is interesting and worth publishing with a slight improvement. In line 63 the authors use the term "supermolecular hydrogels" but it seems that "supramolecular hydrogels" is more appropriate. Loh et al. in ref. 14 cited in the peer-reviewed review also use "supramolecular". Anyway, later in the article, e.g. in Conclusions, the authors already use the term supramolecular.

To address the reviewer’s comment, supermolecular hydrogels was removed and corrected.

Sincerely,

Elham Pishavar

Nanotechnology Research Center, School of Pharmacy

  1. O. Box 91775-1365, Mashhad, IRAN

Tel: +98-513-7112470

Fax: +98-513-7112470

E-mail: Pishavare931@mums.ac.ir

Round 2

Reviewer 1 Report

The authors have thoroughly addressed the entire comments; thus, I recommend accepting the current version of the manuscript.